# The Increasing Investment of Real Estate in the Health System—A Comparison between the USA and Europe

**DOI:** 10.3390/healthcare9121633

**Published:** 2021-11-25

**Authors:** Marta Bachmann, Stephanie Taha-Mehlitz, Vincent Ochs, Daniel. M. Frey, Bassey Enodien, Urs Eriksson, Anas Taha

**Affiliations:** 1Department of Cardiology, University Heart Center, University Hospital Zurich, 8952 Zurich, Switzerland; marta.bachmann@gzo.ch (M.B.); urs.eriksson@gzo.ch (U.E.); 2GZO Regional Health Center, Department of Medicine, Cardiology Division, 8620 Wetzikon, Switzerland; 3Clarunis, University Center for Gastrointestinal and Liver Diseases, 4002 Basel, Switzerland; Stephanie.taha@clarunis.ch; 4Roche Innovation Center Basel, Department of Pharma Research & Early Development, 4070 Basel, Switzerland; vincent.ochs@unibas.ch; 5Department of Surgery, Wetzikon Hospital, 8620 Wetzikon, Switzerland; daniel.frey@gzo.ch (D.M.F.); bassey.enodien@gzo.ch (B.E.); 6Faculty of Medicine, University of Basel, 4001 Basel, Switzerland; 7Department of Biomedical engineering, Faculty of medicine, University of Basel, 4123 Allschwil, Switzerland

**Keywords:** investment, real estate, health system, Europe, USA

## Abstract

Background: This study aimed to compare property development and increasing investment in real estate by the healthcare system organizations in the USA and Europe. Real estate investments have upsurged in healthcare due to the multiple benefits to patients and medical practitioners. Methods: The approach of acquiring data was through secondary sources and online questionnaires. The researchers applied inclusion and exclusion criteria by exclusively including the articles published after 2014 to ensure the validity and reliability of the information. Results: A total of 53.33% of the articles reviewed focused on the United States, while 46.67% concentrated on Europe. The development of real estate in healthcare is essential in both regions due to the challenges faced with the current infrastructure. Study Limitation: Currently, there are very few studies concentrating on the research topic. Conclusions: The USA and Europe should focus on increasing real estate investments in healthcare by focusing on hospitals and trusts, rehabilitation centers, and nursing homes.

## 1. Introduction

### 1.1. Topic Background

Real estate in healthcare represents the buildings, agencies, and campuses rented to individuals or organizations in the healthcare community. The owners of the establishments are usually hospitals, healthcare structures, private entities, and public third-party teams. The development and growing investment of real estate in the healthcare system has led to numerous advantages. For instance, a study by Van der Voordt on the Dutch healthcare system reveals that the transformation of the Netherlands’ healthcare system from a governmentally driven sector to a regulated marketplace force has led medical structures to acquire full responsibility for their real estate [1]. However, this form of investment also triggers some challenges. Gupta et al. point out that the success of private equity in the healthcare system agonizes from exceptional market frictions, as patients cannot directly evaluate provider quality since payments for services are not direct [2].

### 1.2. Problem Statement

The researchers hoped to address the research problem by filling the research gap on developing and increasing real estate investment in the healthcare system. For instance, the study compares the situations between the USA and Europe, which is an understudied area. Currently, very few studies have focused on the topic. In the real estate sector, real estate in healthcare is a niche market [3] that needs further exploration to reap the multiple benefits it encompasses.

### 1.3. Research Rationale

The findings acquired through this study will benefit current real estate and healthcare students by providing them with additional information that they can use as references in their essays on this subject or any other related topics. Moreover, the data gathered via this research presents researchers with information that they can use to enhance their research on similar or associated research. Additionally, researchers can critique the work and provide new findings on the subject. Furthermore, this study could benefit various hospitals, rehabilitation centers, and senior residents’ homes in the USA and Europe by educating them on the current developments and investments in real estate. The governments in the respective regions can use the data gathered through this study to address the challenges that enterprises face when developing real estate for various healthcare institutions.

### 1.4. Research Aim and Questions

The main objective of this study was to analyze and compare the development and increasing investment in real estate in the healthcare system between the USA and Europe. The following questions assisted in attaining this aim:How have hospitals and trusts in the USA and Europe developed and increasingly invested in real estate?How have rehabilitation centers in the USA and Europe developed and invested in real estate?How have senior residents’ homes in the USA and Europe developed and invested in real estate?

### 1.5. Outline

This article comprises four more chapters, where the next section responds to the research questions by evaluating the available literature on the topic. More importantly, the research methodology chapter highlights the approaches adopted by the researcher to acquire data on the subject. The results and analysis of data evaluates the findings derived from the research and presents their implications. Finally, the concluding chapter details the key points discussed in the article.

## 2. Materials and Methods

This manuscript utilizes distinct contributions to handle this complex subject. The methodological approach adopted consisted of theoretical and interactive stages. The utilization of the phases was to acquire data on the increasing investment in real estate in the healthcare systems of the USA and Europe via multiple sources so as to have independent categories of information to compare. During the first phase, the authors analyzed the articles that previous researchers had developed on the topic. The second phase entailed collecting primary data by distributing questionnaires to companies working in real estate and to patients of the different facilities.

### 2.1. Theoretical Stage: Review of Current Literature

This step entailed the identification of previous literature on real estate investment in healthcare systems in the United States and Europe. The researchers conducted a literature search on the PubMed database and the Google Scholar search engine from September to October 2021 to acquire relevant data on the topic. The authors of this manuscript used a variety of keywords to retrieve relevant documents. Among the keywords used were “real estate investment in healthcare”, “real estate investment in hospitals”, “real estate investments in rehabilitation centers”, “real estate investment in senior residents’ homes”, “the United States”, and “Europe”. The search yielded 40 documents, whereby 15 of the documents underwent elimination due to duplication and 5 due to the availability of the document in another language other than English. Further analysis led to the elimination of 5 irrelevant papers to the topic, with fifteen remaining.

### 2.2. Interactive Stage: Online Questionnaires

This phase entailed distributing online questionnaires among individuals from healthcare REITs in the USA and Europe and among different patients receiving general hospital, rehabilitation, and senior care services. The gathering of respondents for the study entailed the researchers directly contacting healthcare REITs through telephone to request the distribution of the questionnaires in the establishments with staff and patients. The first part of the questionnaire entailed the respondents giving general information about themselves and whether they were investors or patients. The other sections of the questionnaire highlighted questions concerning the efficiency of the services delivered by healthcare REITs and the various areas in need of improvement.

### 2.3. Comparison Stage

This step aimed at establishing a comparative analysis of the information gathered from the literature review and the online questionnaires.

## 3. Results

### 3.1. Literature Review

The healthcare systems adopted in the United States and different parts of Europe are distinct. For instance, the number of hospitals in the USA is 6090, while Europe has 24,200 hospitals. Additionally, individuals lack access to public healthcare insurance in the USA, while medical coverage in Europe is universal. Lorenzoni, Belloni, and Sassi point out that the USA has the most significant number of residents without health coverage among all Organisation for Economic Co-operation and Development (OECD) countries [4]. Another difference between the two systems is that the cost of medication and treatment is higher in the USA compared with Europe. For instance, European countries have found methods to successfully negotiate with drug companies to reduce the cost of medicine, whereas the United States continues to battle high drug prices. A study by Van-Overbeeke et al. proved this. It revealed that the price of gene therapy medicinal products is higher in the USA than in countries in the European Union [5]. In the USA, real estate investment in healthcare is an outcome of the region’s medical Real Estate Investment Trusts (REITs). Similarly, healthcare REITs have dominated the European market since the commencement of investments.

#### 3.1.1. The Increasing Investment in Real Estate by Hospitals and Trusts in the USA

An article written by Taylor revealed that the need for healthcare real estate in the United States is increasing [6]. For example, healthcare real estate has plummeted to a value of USD 1.2 trillion [6]. Another study by Rivas et al. showcased that the ZIP Codes in the United States with hospitals demonstrate a below-average median house price and median rent [7]. However, this does not apply to the area codes with large hospitals [7]. Most important, a study initiated by Turchi et al. on the Center for Children and Youth with Special Healthcare Needs demonstrated that real estate investment in the sector has grown [8]. Primary care for the youth and children in the CYSHCN has expanded to various settings, including community health centers, clinics based in hospitals, and private entities [8]. This argument insinuates that many investors aim at developing real estate in hospitals to cater to the needs of sick populations in America.

Additionally, Hamilton and Peiser claimed that historically, the decentralization of land use in the USA resulted from local authorities and boards regulating the majority of the decisions in the sector [9]. Presently, financial incentives geared toward development and direct regulation shape land utilization in the region [9]. Furthermore, Canning argued that the onset of the coronavirus pandemic has necessitated the need to invest in real estate for hospitals and trusts [10]. For example, the authors advise hospitals to replace the commonly used steel door handles with future retrofitting materials that offer improved, low cost, and more diversified anti-viral options compared with what hospitals have today [10]. Another review by Veuger illustrated that the real estate domain had undergone numerous transitions in treatment methods, such as kidney dialysis, chemotherapy, and infusion therapies [11]. Therefore, these treatment approaches will affect the real estate of hospitals and trusts with themes such as price erosion, increased demand, and overcapacity [11], thereby implying that the investment in real estate by hospitals and trusts is likely to increase with time.

#### 3.1.2. The Increasing Investment in Real Estate by Hospitals and Trusts in Europe

The investment in real estate by hospitals located in Europe has increased over the years. For example, research conducted by the Statista Research Department showcased that the investment in healthcare real estate in Western Europe surpassed EUR 7 billion in 2019 [12]. In 2019, Germany recorded the highest investment volumes with EUR 1.68 billion followed by the UK (EUR 1.46 billion), Netherlands (EUR 1.2 billion), Sweden (EUR 1.1 billion), Belgium (EUR 0.5 billion), France (EUR 0.44 billion), Finland (EUR 0.42 billion), Spain (EUR 0.3 billion), and Italy (EUR 0.21 billion) [12]. However, Viergutz argued that German hospitals face the shortcoming of the demographic aging of the population, countryside departure, and the rampant evolution of healthcare technologies [13]. Solving these problems will require the country to establish more hospitals in new locations to meet the demand for hospital services [13]. An article by Breuer and Steininger revealed that Germany ought to undertake a regional analysis of its housing market before developing the healthcare real estate domain. Furthermore, the authors demonstrated that the COVID-19 pandemic would steer the development and investment in real estate of hospitals in Europe, as many healthcare facilities will have to enhance their hygiene, which might require structural in addition to technical transformations [14].

Another study conducted by Wijburg showcased that the increased real estate investment in trusts, also known as REITs, in France has led to significant results [15]. For example, the reconfiguration of the property market in France has permitted the introduction of new regulations and awarding REITs the capacity to participate in matters involving property development [15]. In a similar view, Newell and Marzuki stated that the REITs in Germany have availed sturdy risk-adjustment returns and outperformed other property companies in the region [16]. German REITs significantly contribute to a mixed-asset portfolio, which permits small pension funds to use the trusts to attain property exposure in liquid form [16]. The exquisite performance of REITs implies that investing in them will be beneficial to the community; hence, countries in Europe will continue developing such establishments. In Central Eastern Europe, real estate investment in healthcare has plummeted over the last twenty years since the region has undergone economic transition and a significant period of capital accumulation. As an outcome, the price of investments in the area has become cheaper [17].

#### 3.1.3. The Increasing Investment in Real Estate by Rehabilitation Centers in the USA

Taylor argued that in the United States, healthcare spending will probably grow to 19.7% of GDP by the year 2026, and this sector has a crucial role to play in ensuring better, more accessible, and convenient healthcare services, including rehabilitation offerings [6]. A report published by CBRE Capital Markets in 2019 revealed that 27% of potential investors envisioned inpatient rehabilitation facilities as possible acquisition facilities [18]. More important, the report outlined that many investors involved in the study predicted that the price of rehabilitation hospitals in the USA would range from 6.5% to 7.49%, compared with the prices in 2018 that ranged from 7% to 7.99% [18]. These statistics imply that the reduced prices for rehabilitation centers will steer more investment since the benefits acquired through these establishments will be more than the costs incurred during purchase. This trend is likely to encourage more people to invest in rehabilitation facilities.

A review by Amatya and Khan showcased that the COVID-19 pandemic has necessitated the introduction of more rehabilitation centers in the United States of America, as the pandemic has triggered numerous human, financial, and public expenses [19]. The authors further pointed out that the individuals who survive the pandemic may experience multiple clinical and psychological impairments that require rehabilitation [19]. For instance, mental health issues have increased, particularly in cases where individuals have lost a loved one to COVID-19. Such individuals will need mental health services to prevent the deterioration of survivors’, victims’, and medical professionals’ wellbeing. In a similar perspective, Lugo-Agudelo et al. argued that the lingering requirement for rehabilitation centers stems from the inadequacy of services received by patients with or without coronavirus illness [20]. Therefore, this phrase implies that more people will invest in rehabilitation centers to ensure that patients receive the necessary care and do not succumb to the challenges they face amid the pandemic.

#### 3.1.4. The Increasing Investment in Real Estate by Rehabilitation Centers in Europe

In Europe, the need for real estate investment in rehabilitation centers has plummeted in Germany, particularly during the coronavirus pandemic. For instance, an article by Bauer states that the investment in rehabilitation clinics has increased due to tremendously high demand in Germany [21]. The expanded demand results from demographic shifts and the increment in chronic illnesses and surgeries that require consequent inpatient rehabilitation [21]. These statements signify that the investment in rehab centers is likely to increase to cater to the increasing needs of the population. Furthermore, the World Health Organization report showcased that the European continent needs rehabilitation services due to the COVID-19 pandemic [22]. The organization further insisted on the lack of healthcare professionals to provide rehabilitation therapy to survivors of the pandemic [22]. Based on this argument, the development of rehabilitation centers is likely to increase after the pandemic to prevent the adverse effects experienced if another pandemic occurs.

Another study spearheaded by Bayly et al. revealed that the pandemic significantly disrupted rehabilitation services in Europe [23]. Despite the adverse outcomes created by the situation, healthcare professionals and investors can utilize the instance to steer positive changes in the delivery of the offerings to patients [24]. Based on this argument, real estate investment in rehabilitation centers remains crucial in improving health outcomes among the European population. Additionally, Aguiar de Sousa highlighted a survey by the European Stroke Association that showcased that rehabilitation care for patients with stroke was poor during the pandemic, mainly due to organizational changes [24]. For instance, the COVID-19 pandemic forced many healthcare professionals in Europe to adapt their schedules. In contrast, half of the surveyed respondents stated that they lacked sufficient protective equipment to deliver care to patients [24]. In other words, the pandemic is likely to steer investments in rehabilitation centers. In Central Eastern Europe, the demand for old houses will decrease due to the preference for more modern establishments [25]. Therefore, investors will likely construct new healthcare centers that have current services, inclusive of rehabilitation centers.

#### 3.1.5. The Increasing Investment in Real Estate by Senior Residents’ Homes in the USA

A report carried out by Saiz and Salazar revealed that the increasing acquisition of real estate by senior residents’ homes in the USA is likely to go up since the aging of the American residents is an inevitable reality that requires the country to plan well and ensure the housing of the elderly [26]. For instance, the USA can guarantee this by retrofitting the existing residential homes and uptown spaces [26]. Taylor further supports this argument by stating that the growth of the United States population is likely to upsurge by roughly seventy-nine million individuals by the end of the year 2060 [6]. Gupta et al. pointed out that private equity in nursing homes in the United States has made the facilities insist on profits and short-term returns, thereby reducing the quality of care delivered to residents [2]. This statement implies that more publicly funded investments are necessary to guarantee that nursing homes do not exclusively focus on profits.

A study by Abrams et al. revealed that the pandemic has negatively affected nursing home residents’ by causing massive death, as the occupants were elderly and at a higher risk of suffering the consequences of the illness [27]. The study’s findings revealed that the pandemic outbreak in USA nursing homes was significantly a result of the size and location of the facility [27]. Therefore, this fact insinuates that the country needs to construct increased and more prominent senior residents’ homes in different areas through real estate investment to prevent such instances if another outbreak occurs. Furthermore, a commentary by Fulmer, Koller, and Rowe demonstrated that over 65% of nursing homes in America are for-profit, while the caregivers are often poorly trained and paid [28]. Despite the challenges, the pandemic provides a new chance for the American healthcare system to reimagine the responsibilities of nursing homes and how real estate investors can redesign future nursing homes to cater to the requirements of residents and healthcare providers [28].

#### 3.1.6. The Increasing Investment in Real Estate by Senior Residents’ Homes in Europe

The investment in real estate in nursing homes in Europe has grown, given the region’s increasing population of elderly individuals. A study by Huisman et al. revealed that decision makers in the area aim at improving the life quality and satisfaction of individuals in nursing homes by including residents and staff in the decision-making processes of increasing investments in real estate [29]. Kazak et al. argued that countries in Western Europe had begun raising the old-age category from 65 to ensure that the older generations engage in communal and societal activities while having improved physical health compared with previous generations [30]. The care required by the elderly is complex and requires European countries to continuously invest in national housing systems to take care of them [30]. Gupta et al. insist that private equity nursing homes in the United Kingdom aim at making profits and cutting costs and that the owners load up the establishments with debts and massive interest rates [2].

Moreover, a report released by Statista Research Department showcased that Germany had 23.1% of all real estate investment in care homes in Western European in 2019 [27]. They were followed by the UK (20%), Netherlands (16.5%), Sweden (14.8%), Belgium (6.7%), France (6%), Finland (5.7%), and Spain (4.1%) [31]. The report further revealed that Italy had the lowest investment volume in nursing homes at 2.89% [31]. A review by Bos, Kruse, and Jeurissen showcased that for-profit senior resident homes are rising in the Netherlands [32]. The primary reasons for this change are the transitions in the region’s regulatory framework and the low responsiveness of non-profit entities to the changing demands of nursing homes, and an increase in the number of private equity organizations [32]. Another study by Deusdad highlighted that Europe had showcased weaknesses in the country when handling health crises such as the current COVID-19 pandemic due to the tendency to neglect the wellbeing of older generations [33]. In Spain, the pandemic has necessitated the need to ensure older people’s rights, but this has encountered a myriad of shortcomings associated with resource scarcity in nursing centers [33].

The articles reviewed for this research were fifteen, whereby eight of them (53.33%) concentrated on the United States, whereas seven (46.67%) availed data on the European continent. One of the articles included in the study evaluated the prevalence of nursing homes in the United States and Europe and was mentioned twice in the review. From the analysis, it remains evident that both countries need to increase their real estate investment in healthcare to deal with the challenges of the current buildings. Additionally, the studies reveal that the COVID-19 pandemic will steer a transformation toward more investments to reduce congestion in various healthcare facilities. Table 1 details the distinct themes of the articles.

### 3.2. Online Questionnaire

The researchers distributed questionnaires to twenty individuals, out of which nine people replied to the questionnaire. Out of the nine, five represented the investor, while four of the respondents expressed patients’ views. Of the four individuals, three received general hospital services, while one person received rehabilitation offerings. None of the respondents represented the senior residents’ domain.

## 4. Discussion and Analysis

The investment in real estate by the healthcare system in the United States of America and Europe has increased remarkably during the pandemic. However, there is a lack of sufficient studies on the topic, thereby presenting a research gap in the area. Nevertheless, both continents have invested in Real Estate Investment Trusts over the years to increase the quality of care for patients. For instance, Akinsomi stated that the market capitalization for REITs in the United States was at 48.32%, which was the highest [34]. Furthermore, Adnan et al. pointed out that the significant REITs in the healthcare system in the United States consist of Health Care REIT Inc. and HCP Inc., in addition to Ventas Inc. [35]. This factor implies that the country is likely to advance investments in REITs even after the pandemic. In a similar perspective, Newell and Marzuki associate REITs with the enhanced performance of healthcare trusts in Germany [16]. Therefore, the high prevalence of REITs demonstrates that real estate investors will probably invest in the sector in both locations to reap the associated benefits.

Another similarity between the two regions is that the COVID-19 pandemic has increased the need for the areas to invest in rehabilitation centers. For example, Amatya and Khan insisted that the psychological impacts of the pandemic make it necessary to provide rehabilitative services to both healthcare professionals and survivors in the United States [16]. Furthermore, Lew, Oh-Park, and Cifu insisted that the pandemic significantly affected the USA [36]. The most appropriate way to minimize the complication of the disease was by providing interdisciplinary rehabilitation to patients and professionals that meets the various needs of the population [36]. Moreover, Bayly et al. pointed out that the pandemic disrupted the delivery of rehabilitation services in Europe and simultaneously presented new opportunities for future development [23]. Stucki et al. stated that the World Health Organization recognizes the establishment of rehabilitation centers as among the primary goals for 2030 due to its ability to drive societal returns [37]. Additionally, the authors highlighted the need to classify rehabilitation services as public health strategies under the UHC [37].

Both the United States of America and Europe showcased a prevalence of private equity investments in nursing homes guided by the need for profit [2]. Moreover, both countries have a prevailing number of elderly individuals with a myriad of diversified needs that care homes provide. In the USA, the need to retrofit existing homes to meet the increasing demands of the elderly population is essential for ensuring quality care [25]. In Europe, the main reason why real estate investors should consider nursing homes in the region is the tendency of the area to ignore the needs of older individuals, especially during pandemics [29]. Therefore, these statistics imply that both localities will increase real estate development in healthcare to deal with the lingering challenges in nursing homes by developing establishments that do not focus on profits but deliver quality care to older people. However, the healthcare systems adopted in the United States and Europe are different, implying that healthcare systems’ levels and efficiency will differ. It remains critical for investors to evaluate both markets well before investing in healthcare real estate.

## 5. Conclusions

Real estate in healthcare represents the establishments leased to individuals in the healthcare community, where hospitals or public third-party teams typically own the establishments. The real estate healthcare sector has been underestimated and requires more exploitation to incur more profits. The information in the study will assist various enterprises, including the government, in preventing the challenges that come with the development of real estate in the healthcare sector. Since aging is inevitable, America will require more investment in the healthcare real estate sector in senior individuals’ residences.

## 6. Limitations

Nonetheless, the main limitation of this study is the lack of sufficient studies published in the topic area. Additionally, the inclusion of only English articles could have prevented the researchers from gathering information in other languages. Finally, the online questionnaire results remain biased since they did not include the views of the elderly, who are the primary users of care homes. The study findings reveal that there are massive challenges in the healthcare systems of the USA. For example, healthcare is not free, and the cost of medicines is higher compared with Europe. In Europe, the current developments in CEE signify that investments in healthcare facilities will spiral in the next years. Evaluating the subject from a non-market perspective reveals that the demand for hospital services has increased as more people get sick and age. Nonetheless, both countries can survive the shortcomings of the pandemic, given that they adopt the recovery plans incorporated. The USA integrated a Goal 7 plan, while Europe adopted the Recovery Plans for EU Members. The successful execution of these plans will make the countries more prepared for future pandemics and will create room for real estate investment in medical systems.

## Figures and Tables

**Table 1 healthcare-09-01633-t001:** Comparison of the USA and Europe real estate investment in healthcare.

Article	Main Theme
USA	
Akinsomi [34]	The market capitalization of REITs is highest in the United States.
Amatya and Khan [16]	Psychological impacts of the pandemic necessitate the need for rehabilitation centers.
Gupta et al. [2]	Private equity investments are rampant and guided by profits.
Adnan et al. [35]	Prominent REITs in the healthcare domain are Health Care REIT Inc., HCP Inc., and Ventas Inc.
Lew, Oh-Park, and Cifu [36]	Interdisciplinary rehabilitation is critical in the region.
Saiz and Salazar [26]	Retrofitting nursing homes is necessary to meet the needs of residents.
Lorenzoni, Belloni, Sassi [4]	High cost of healthcare in the US.
van Overbeeke et al. [5]	Gene therapy medication is more costly in the USA that Europe.
Europe	
Newell and Marzuki [16]	REITs have promoted the increased performance of healthcare trusts in Germany.
Bubbico et al. [17]	Market transitions in Central Eastern Europe encourage real estate investments.
Bayly et al. [23]	The pandemic disrupted rehabilitation but offered opportunities for improvement.
Gupta et al. [2]	Private equity investments in senior residents’ homes based on profits.
Stucki et al. [37]	WHO envisions the development of rehabilitation centers as a key goal.
Deusdad [33]	The nursing homes domain significantly ignores the wellbeing of the elderly.
Rącka, Palicki, and Kostov [25]	The demand for new buildings in CEE have resulted in a drop in the cost of real estate investments.

## Data Availability

The datasets used and/or analyzed during the current study are available from the corresponding author on reasonable request.

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
