# Peer review of "The Increasing Investment of Real Estate in the Health System—A Comparison between the USA and Europe"

_healthcare, 2021, doi:10.3390/healthcare9121633_

Round 1
Reviewer 1 Report
The title of the reviewed article raises two types of doubts. First, according to the opening part: ‘The development and increasing investment of real estate in the health system’. After all, the Authors say twice about the same. Developing real estate investments in the health system and increasing real estate investments in the health system are, in fact, unnecessary repetition of wording. It seems that, taking into account the real content of the paper, it is about diagnosing the needs for the development of various types of medical facilities and about the reasons, conditions and goals of investing in such real estate. The title of the article should oscillate around these issues. The second part of the title of the study proposed by the authors (i.e. ‘A comparison between the USA and Europe’) suggests that the conditions, causes, needs, symptoms and goals of investing in medical real estate comparing the USA and Europe may be at least partially different. Undertaking a comparative analysis as a research tool is rational only in the case of noticing certain differences in the research objects. Otherwise, it loses its sense - it does not result in any added value, which is what science is all about. Meanwhile, the analysis clearly shows that, according to the Authors, both regions (USA and Europe) struggled with the same problems, conditions, and needs in terms of the functioning and development of health care facilities. The conclusions drawn from the research are basically the same in the context of the USA and Europe.
The topics undertaken by the Authors are generally interesting, but the research approach is extremely simplified and fragmentary. They do not show the differencies in conditions, specificity of the functioning of health care systems in the USA and parts (countries) of Europe. There is no diagnosis of the initial situation, no strategic assessment of the phenomena that embed the functioning of medical properties in both compared geographical units. After reading the text, it seems that despite the time scope of the research defined as 2014-2021, it is devoted almost exclusively to the effects of the covid pandemic, and thus the situation (the surroundings of the analyzed area, its conditions, needs, problems) from 2019 is assessed.
The title of the article suggests the USA and Europe as a geographic scope, while the entire text completely omits Central-Eastern Europe (CEE). This area is interesting and inspiring because in the last 20 years it has become an arena of the most dynamic changes in the analyzed issues. The spontaneous development of market mechanisms in the post-socialist countries has crystallized new ownership and management models in the development of medical facilities. The different conditions in this part of Europe would certainly allow the Authors of the article to draw interesting conclusions and would justify the concept of comparing Europe and the USA.
The entire article has been prepared by six scientists. In the empirical layer it focuses on the analysis of 10 publications on a strictly discussed topic. With such a small representation of the analyzed texts and the reliance solely on secondary data, it is difficult to identify the order (the pattern) in complex phenomena and properly conclude about them. Rather, these are fragmentary considerations based on isolated examples. They provide a simple overview of the case studies while it should be based on a systemic approach.
The Authors incorrectly calculate even the share of the analyzed articles about the USA and Europe. Although there are 10 articles in total, if one of them refers to the USA and Europe at the same time, it should be counted twice (treating 11 elements as the sum of the set of examined articles). Thus, the actual shares of articles relating to the USA topic constitute 54.5% of the total, and the shares of articles relating to Europe constitute 45.5% of the total (and not, as the Authors say, 60% and 40% respectively). Incidentally, it seems that the co-authorship of as many as six scientists involved in the preparation of the reviewed paper is a really large number.
Finally, the conclusions drawn by the Authors in the article are superficial and obvious. It is not surprising at all that from the point of view of investors the most important factor is the financial efficiency of implemented projects in the commercial and private model as the dominant formula for organizing, financing and managing investments in health care facilities. This again shows the need to analyze the situation in the ‘non-market’, public formula of organizing similar facilities (and such examples are provided by both the USA and Europe, including CEE). It is also clear that the covid pandemic has sharply increased the pressure on the healthcare system on both sides of the Atlantic Ocean, but also around the world.
In the entire reviewed study, the methodological part deserves praise, in which the Authors correctly argued the choice of the method of determining their research activities. Nevertheless, the actions taken should only be part of a broader analysis that was not included in the article. I recommend supplementing the concept with own research, collecting primary data, e.g. among entities investing in medical real estate, but also among patients of such facilities. The Authors mainly (perhaps unconsciously) adopted the perspective of the supply side, meaning investors in the real estate market, and not necessarily their users, creating demand in this market.
Summing up, I believe that the article requires not only a thorough reconstruction and supplementation. In my opinion, this paper needs to be reformulated primarily in terms of the conceptual level. The article is not suitable for publication in its present form.
Author Response
We changed the topic of the study to the increasing investment of real estate in the health system. We also included an evaluation of the differences in the heathcare systems adopted in the USA and in Europe. We further included an evaluation of the Central-Eastern Europe region. Furthermore, we changed the scope of the study to include primary data to supplement the minimal articles used. We also edited the conclusion to encompass a non-market evaluation of the topic.

Reviewer 2 Report
The paper presents a very interesting topic. aims to compare the development and increasing investment of 12
real estate in the healthcare system of The USA and Europe. The analysis aims to search for pre-existing works and data without adding any originality to the theme dealt with.
Author Response
We added originality to the study by outlining the methodological approach undertaken to gather the data as well as conducting primary research.

Reviewer 3 Report
The paper aims to analyze and compare the development and increasing investment of real estate in the healthcare system between the USA and Europe. The authors explored three types of settings: hospitals, rehabilitation centers and senior residents’ homes. The main conclusion is that investments of real estate in the healthcare sector has increased and it is going to increase due to the COVID-19 pandemic and the ageing population.
The paper does not require extensive changes.
The authors could add some data regarding the state of the art of the healthcare systems in Europe and USA (e.g. number of hospitals and/or other facilities).
The authors could also add a brief overview of international initiatives to provide financial support to the healthcare sector as recovery packages from the pandemic. This could be useful to discuss the future advancements of the healthcare real estate. Recovery Plans of EU member states are an example.
Author Response
We included the number of hospitals in the US and Europe and briefly highlighted the recovery plans adopted by the US and Europe

Round 2
Reviewer 1 Report
The article after changes already made by Authors is worth publishing.
Reviewer 2 Report
The paper has been enriched following my indications. In this case, I can accept the paper